# Systems Biology Analysis of the Antagonizing Effects of HIV-1 Tat Expression in the Brain over Transcriptional Changes Caused by Methamphetamine Sensitization

**DOI:** 10.3390/v12040426

**Published:** 2020-04-09

**Authors:** Liana V. Basova, James P. Kesby, Marcus Kaul, Svetlana Semenova, Maria Cecilia Garibaldi Marcondes

**Affiliations:** 1San Diego Biomedical Research Institute, San Diego, CA 92121, USA; lbasova@sdbri.org; 2Department of Psychiatry, School of Medicine, University of California San Diego, La Jolla, CA 92093, USA; j.kesby@uq.edu.au (J.P.K.); svetlana.semenova36@gmail.com (S.S.); 3Queensland Brain Institute, The University of Queensland, St. Lucia, QLD 4072, Australia; 4Centre for Clinical Research, Faculty of Medicine, The University of Queensland, Herston, QLD 4029, Australia; 5School of Medicine, Department of Biomedical Sciences, University of California Riverside, Riverside, CA 92521, USA; marcus.kaul@medsch.ucr.edu; 6Parexel International, Glendale, CA 91206, USA

**Keywords:** transgenic Tat mouse, methamphetamine, transcriptome

## Abstract

Methamphetamine (Meth) abuse is common among humans with immunodeficiency virus (HIV). The HIV-1 regulatory protein, trans-activator of transcription (Tat), has been described to induce changes in brain gene transcription that can result in impaired reward circuitry, as well as in inflammatory processes. In transgenic mice with doxycycline-induced Tat protein expression in the brain, i.e., a mouse model of neuroHIV, we tested global gene expression patterns induced by Meth sensitization. Meth-induced locomotor sensitization included repeated daily Meth or saline injections for seven days and Meth challenge after a seven-day abstinence period. Brain samples were collected 30 min after the Meth challenge. We investigated global gene expression changes in the caudate putamen, an area with relevance in behavior and HIV pathogenesis, and performed pathway and transcriptional factor usage predictions using systems biology strategies. We found that Tat expression alone had a very limited impact in gene transcription after the Meth challenge. In contrast, Meth-induced sensitization in the absence of Tat induced a global suppression of gene transcription. Interestingly, the interaction between Tat and Meth broadly prevented the Meth-induced global transcriptional suppression, by maintaining regulation pathways, and resulting in gene expression profiles that were more similar to the controls. Pathways associated with mitochondrial health, initiation of transcription and translation, as well as with epigenetic control, were heavily affected by Meth, and by its interaction with Tat in anti-directional ways. A series of systems strategies have predicted several components impacted by these interactions, including mitochondrial pathways, mTOR/RICTOR, AP-1 transcription factor, and eukaryotic initiation factors involved in transcription and translation. In spite of the antagonizing effects of Tat, a few genes identified in relevant gene networks remained downregulated, such as sirtuin 1, and the amyloid precursor protein (APP). In conclusion, Tat expression in the brain had a low acute transcriptional impact but strongly interacted with Meth sensitization, to modify effects in the global transcriptome.

## 1. Introduction

Human immunodeficiency virus (HIV) infection remains a modern challenge and in United States there over one million people infected who are living longer due to treatments with antiretroviral drugs [1]. However, the incidence of neurological disorders associated with HIV infections remains high and is aggravated by comorbidities, such as substance use disorders [2]. The interaction between the virus and its host cells is a combination of transcriptional regulatory mechanisms that benefit the virus, but also affect host gene transcription. The presence of comorbidities such as drug abuse, provides an increased complexity, which needs to be understood in order to address an increasingly high segment of the HIV-infected population. The identification of gene expression signatures through transcriptional profiling and systems biology strategies empower such an understanding and offer mechanistic insights for novel hypotheses.

Of all the drug abuse comorbidities, methamphetamine (Meth) use is one of the most prevalent among humans infected with HIV [3,4,5]. The impact of neurotoxic effects of both Meth and HIV on the brain are well documented [6,7]. However, studies are limited on brain adaptations during the early stages of Meth use and interaction with HIV infection. Modeling HIV neurological disease, and especially the effects of the interaction between HIV and substance use, has been a challenge due to logistical and technical difficulties associated with studying humans, particularly Meth users. Several small animal models have attempted to mimic specific aspects of neuroHIV by driving the expression of the virus or its proteins, or by using viral constructs that can infect mice [8,9,10,11,12,13,14], which all together support the notion that HIV viral products contribute to neuropathology, and affect reward deficits and drug dependence [15,16,17]. The potent effects of HIV proteins in vitro, particularly gp120 and the trans-activator of transcription (Tat), have driven the rationale for the development of small rodent models expressing either one of these proteins, or all HIV proteins, throughout the body or with expression restricted to the brain [5,18,19,20,21]. No small animal model has been able to fully replicate the effects of active infection; most have limitations regarding the significance of the cells expressing the protein, molecular mechanisms, and side effects. The ability to isolate components of the infection argues for the value of each model, especially for examining specific pathways leading to neurological dysfunction. The conditional induction of HIV proteins in brain cells, such as in the doxycycline-driven expression of HIV Tat in astrocytes [18], is a mouse model that could be able to time neurotoxic insults, and mimic the early phases of HIV in the central nervous system (CNS).

Evidence of the presence of viral Tat protein in the CNS of HIV+ human subjects has mostly been indirect and based on the detection of Tat mRNA in the brain of infected subjects [22], the detection of Tat-specific antibodies in cerebrospinal fluid (CSF) [23,24], or of CD4 T cell response patterns that are in agreement with the effects of Tat in vitro [25]. It has been reported that throughout the chronic phase of an infection with the simian immunodeficiency virus (SIV) in rhesus macaques, anti-Tat CD8 T cells persist in the brain but not in the periphery [26,27]. We have also found that Tat-independent mechanisms perpetuate an early response to these proteins [27], additionally, Tat becomes elevated in the CSF of infected subjects [25]. Furthermore, there is evidence of Tat secretion [28], with effects activating and modifying host cells, including innate immune cells, neurons, and astrocytes [29,30,31,32]. We have shown that Tat acts on host cells in vitro, by activating genes involved in inflammation [29]. The effects of Tat in vivo, and its interactions with Meth exposure, can be tested in mice expressing Tat in the CNS to model neuroHIV [18].

Transgenic mice that express the Tat protein, in the brain, under the glial fibrillary acidic protein (GFAP) promoter, and inducible by treatment with doxycycline, show similar signs of neuropathology to what is observed in HIV-infected humans, including gliosis, neuroinflammation, and neuronal loss [18,33,34], and therefore paralleling aspects of neuroHIV. Tat expression in these Tat transgenic animals’ brains was determined to be in the range of 1–5 ng/mL, leading to homogeneous levels of astrocytosis [33]. This model has been instrumental in behavioral studies involving drug-induced sensitization, with Tat-induced dysfunctions in dopaminergic neurotransmission [35,36,37,38,39,40] which can lead to alterations in reward function [36,41,42].

In this present study, we have used the inducible Tat transgenic mouse model to investigate if and how HIV-1 Tat expression in the brain, which impacts Meth-induced locomotor sensitization and dopaminergic responses [43], disturbs more global gene expression patterns following the Meth challenge. The sensitization paradigm has been extensively studied under the premise that early exposures to drugs of abuse increase the magnitude of later drug-stimulated responses [44], with significant consequences to locomotor behavior and the dopaminergic system, which we described in [43]. During the acquisition phase, the animals received repetitive low-dose injections of Meth or saline, followed by a washout period, and then a challenge with the drug. Typically, the animals that received Meth during the acquisition phase had a peak of high locomotor activity after the challenge, while the animals that received saline during the acquisition phase did not show increased motor behaviors at the Meth challenge, as previously described in the same animals examined in [43]. Importantly, animals that expressed Tat in the brain and that received Meth during the acquisition phase had a significantly higher peak of locomotor activity as compared with animals that did not express Tat [43].

We hypothesized that, as in humans and in vitro, the interaction between Tat expression and Meth locomotor sensitization would cause the activation of a large number of genes that could contribute to neurological disease and inflammation. To test this hypothesis, we used our previous experience using systems biology approaches, to conduct an overview of gene expression patterns in the brains of Meth-sensitized animals challenged with Meth or Saline, and on the induction of HIV Tat protein expression in the CNS. After withdrawal, all animals were challenged with the drug. As described previously, Meth sensitization caused an increase in locomotor activity, and this was further increased in animals expressing Tat, while saline sensitization did not have that effect [43]. We have focused our analysis on the caudate putamen that is a region with dopaminergic projections which, in humans, is significantly impacted by HIV and also by drug abuse [45,46,47]. Our results challenged the original hypothesis, showing that the impact of Tat expression in the context of Meth sensitization resulted in mild global transcriptional changes and signatures. We analyzed these changes in detail in order to estimate the impact of Tat alone, of Meth sensitization alone, and of their interaction, and the implications of stronger signatures to pathogenesis. The results provided a different perspective on the effects of Meth and the viral protein Tat on transcriptional responses caused by Meth.

## 2. Materials and Methods

### 2.1. Animals

We tested a total of 20 male mice (3 to 5 months old), i.e., 10 containing the GFAP promotor-controlled tetracycline (Tet)-binding protein (Tat−) and 10 containing both the GFAP promotor-controlled Tet-binding protein and the (tetracycline responsive element) TRE promotor-Tat protein transgene (Tat+). Inducible Tat transgenic mouse colonies with a C57BL/6J background were obtained by the generation of two separate transgenic lines, Teton-GFAP mice and TRE-Tat 86 mice, and then cross-breeding of these two transgenic mouse lines, as previously described [18]. The mice were housed in groups of 2 to 4 in a humidity- and temperature-controlled animal facility on a 12/12 h reverse light/dark cycle (lights off at 7 a.m.) with ad libitum access to food and water. All experiments were conducted in accordance with the guidelines of the American Association for the Accreditation of Laboratory Animal Care and National Research Council’s Guide for the Care and Use of Laboratory Animals and approved by the University of California San Diego and San Diego Biomedical Research Institute Institutional Animal Care and Use Committees (BHR-17-001).

### 2.2. Doxycycline Regimen

All mice, 10 containing the GFAP promotor-controlled tetracycline (Tet)-binding protein (Tat-) and 10 containing both the GFAP promotor-controlled Tet-binding protein and the (tetracycline responsive element) TRE promotor-Tat protein transgene (Tat+), were treated with a doxycycline regimen (doxycycline hyclate; Sigma, St. Louis, MO, USA) of 100 mg/kg, intraperitoneally, once a day for 7 days (after locomotor testing), beginning in the evening (5 PM) before the Meth acquisition phase. This regimen was based on the previously demonstrated efficacy of Tat induction of doxycycline at this dose [48,49]. Tat expression was significantly attenuated 14 days after the termination of doxycycline treatment [49]. Only mice containing both the GFAP promotor-controlled Tet-binding protein and the TRE promotor-Tat protein transgene (Tat+) generated Tat protein after doxycycline administration.

### 2.3. Methamphetamine-Induced Sensitization

All the mice underwent the Meth or saline-induced sensitization procedure and were tested behaviorally for locomotor activity, which has been published [43].

The administration procedure consisted of seven consecutive days of intraperitoneal injection with either saline (0.9% Sal) or 2 mg/kg Meth (methamphetamine hydrochloride; Sigma), with doxycycline administered at least 1 h after locomotor testing. The Meth doses were selected based on the literature [50,51]. There were four experimental groups, with 5 mice per group, as follows: (1) Tat-negative (−)/Sal, (2) Tat+/Sal, (3) Tat−/Meth, and (4) Tat+/Meth. After a 7-day abstinence period, all animals were challenged with a single dose of 1 mg/kg of Meth, and brains were harvested for analysis 1 h after the challenge.

### 2.4. Brain Harvest

Mice were euthanized 30 min after completing the locomotor challenge (1 h after injections of Meth). Following perfusion with ice-cold PBS containing 0.2% EDTA, brain samples were rapidly dissected, and samples frozen on dry ice and stored at −80 °C until analysis. The dissected caudate-putamen region was used for microarray and quantitative polymerase chain reactions (qPCRs). Other dissected regions, such as the nucleus accumbens and ventral tegumental area, were used for neurochemical and molecular analyses, and the results for that have been published [43].

### 2.5. Gene Expression Array

RNA from the brain was extracted from the caudate putamen using a Qiagen RNeasy Mini kit (Qiagen, Germantown, MD, USA). The integrity of the extracted RNA and the total RNA concentrations were examined in an Agilent Bioanalyzer 2100 (Agilent Technologies, Santa Clara, CA, USA). The global expression of genes was measured using the Agilent microarray service, performed by Phalanx Biotech (San Diego, CA, USA). RNA was labelled using a Cy5 dye Turbolabelling kit (Thermofisher Scientific, Waltham, MA, USA) following the manufacturer’s instructions. A total of 4 µg Cy5-labeled RNA targets were hybridized to Gene Expression v2 4 × 44K Microarrays (Agilent Technologies, Santa Clara, CA, USA), and analyzed according to the manufacturer’s protocol. Following the hybridization, fluorescent signals were scanned using an Axon 4000 (Molecular Devices, Sunnyvale, CA, USA). Five replicates per condition were used. Microarray signal intensity of each spot was analyzed using the GenePix 4.1 software (Molecular Devices, Sunnyvale, CA, USA). Each signal value was normalized using the R program in Limma linear models package (Bioconductor 3.2, https://bioconductor.org).

Gene expression was calculated by loading raw data into ArrayStudio (Omicsoft Corporation, Cary, NC, USA), with the first filter based on a built-in analysis of variance (ANOVA), as well as a *t*-test, applied to fold changes between experimental and control conditions. Significant changes had a *p* value < 0.05. In addition, maximum least-squares (Max LS) mean of 6 and a false discovery rate by the Benjamini–Hochberg correction (FDR_BH) of <0.01 were applied. Using this method, genes that were found with raw *p* values < 0.05 and where the FDR_BH did not reach <0.01 were discarded. The genes were further filtered to express a robust above or below 3-fold significance, above background, gene expression change.

### 2.6. Systems Approach

Principal component analysis (PCA) and hierarchic clustering of total gene expression data were performed using BioVinci (Biotouring, San Diego, CA, USA). Comparisons between assigned experimental groups were performed with a cutoff on two-fold expression changes, up or down. Pathways and processes with overrepresentation in assigned group comparisons were identified based on Z-score transformation [52,53], calculated from the raw values in Ingenuity Pathway Analysis (IPA) (Qiagen Bioinformatics, Redwood City, CA, USA). These transformed values were saved in .xls and .txt formats to be used as an attribute list for visualization. Gene network interactions were identified using Genemania [54] as an application in Cytoscape 3.7 (National Resource for Network Biology, National Institute of Health, USA, http://www.cytoscape.org) [55], and confirmed using Biogrid *Mus musculus* version 2017-07-13 interaction database [56] as a base network. Interactions identified in Genemania were filtered to include pathway, physical and genetic interactions, and shared protein domains in edges attributes. Functional annotations were verified by analyzing networks using BINGO plugin [57], uploaded into Cytoscape as application, which provides access to Gene Ontology (GO) terms (http://geneontology.org/page/go-enrichment-analysis) databases. Transcriptional factor usage predictions associated with significantly changed genes were performed by the following two ways: (1) by applying the Match algorithm in TRANSFAC 4.0 [58,59] (http://www.portal.geneXplain.com/TRANSFAC) or (2) by using iRegulon [60] as a plugin application in Cytoscape. The use of each method is described in the text of the Results Section. The TRANSFAC database was accessed using a licensed computer for predicting transcription factor binding motifs frequency, and building nucleotide positional weight matrices, searched in the complete dataset, and in one-by-one comparisons assembled in individual .xlm documents, and containing identifiers such as gene name, and accession numbers. The data was loaded into the gene-level microarray feature, selecting *Mus musculus*, with EntrezGeneID as the identifier, fold changes, and FDR_BH *p* values as observations. Fold change observations that were upregulated were cut off to above 3-fold, with background selected as the non-change set from the same experiment set. Default analysis parameters were pre-grouped vertebrate non-redundant matrices, with selected TRANSFAC database version 2018.3, *p* < 0.01, with a false positive minimizing setup, and high-quality matching within −500 and +100 binding pairs relative to the transcription starting site. Matrix summaries and sequence details were exported and saved as .txt. Following the identification of significant transcription factor motifs based on high score matrices, a reverse analysis of target genes was performed using the TRANSFAC-based custom build search for genes exhibiting the identified individual and combined transcriptional factor binding motifs, which generated a list that was further examined in Pathfinder link, and visualized in Cytoscape, using Genemania plugin. In addition, target-gene networks were generated in iRegulon for each of the most significant predicted transcription factors, and merged redundancy detection. Annotations in .txt format were loaded into these networks for visualization of the effect of different conditions. Figure captions describe visualizing codes.

### 2.7. RT-PCR

Gene expression validations of interest were performed on RNA from biological replicates, extracted using Qiagen kits, and reverse transcribed using SuperScript III Reverse Transcriptase (Invitrogen, Waltham, MA, USA). Primers for RT2 qPCR were purchased from Qiagen (Valencia, CA, USA), for SIRT1 (catalog number PPH02188A-200) and APP (catalog number PPH05947A-200). PCRs were performed using RT² SYBR Green ROX FAST Mastermix (Qiagen) in a 7900HT Fast Real-Time PCR System with Fast 96-Well Block Module (Applied Biosystems, Foster City, CA, USA) with an SDS Plate utility v2.2 software (Applied Biosystems). The results were normalized to the geometric mean of both glyceraldehyde 3-phosphate dehydrogenase (GAPDH) and 18S ribosomal RNA housekeeping genes, with PrimeTime qPCR assay oligomers (catalog numbers Hs.PT.39a.22214836, and Hs.PT.39a.22214856, respectively, Integrated DNA Technologies, San Diego, CA, USA).

### 2.8. Statistical Analyses

Global transcriptional volume was estimated by the sum of all fold changes in Excel platform. Data were analyzed in IBM SPSS Statistics 20 using ANOVA, with Tat, Meth, and interactions, as well as the between-subject factors, (Armonk, NY, USA). The a priori hypothesis was that Meth alone would have a large effect as compared with Tat expression, but the combination of both would result in the largest magnitude of change. When appropriate, post hoc comparisons were performed using Bonferroni’s test. Differences were considered statistically significant at *p* < 0.05.

## 3. Results

### 3.1. General Findings

The transcriptional profiles in the brains were characterized using an Agilent mouse gene expression platform. Critical genes were validated by qPCR, in the mRNA extracts from animals that were Tat− and received saline as the vehicle during the acquisition phase (Tat−/Sal), Tat− and received Meth (Tat−/Meth), Tat+ and received saline (Tat+/Sal), and Tat+ and received Meth (Tat+/Meth), as described in methods.

Group differences in gene expression patterns were filtered by the FDR_BH *p* value, with alpha 0.05. This revealed signature genes that characterize the effects of Tat expression, of Meth, and of their interaction on gene expression. The Tat−/Meth mice were dramatically different from the other groups (Figure 1); these were characterized by an overall suppression of gene expression patterns normalized to the average of combined probes, with segregation visualized in a heat map (Figure 1A), and confirmed by the calculation of global transcriptional volume based on the sum of all group-averaged fold changes (Figure 1B), and by PCA (Figure 1C), without excluding outliers. The Tat−/Meth had the largest impact on gene expression, whereas in the animals that were Tat+/Meth, the overall expression pattern was closer to controls Tat+/Sal or Tat−/Sal. A two-factor general linear model with Tat and Meth *t*-tests and *F*-tests was performed to show the contrasts and identify signatures. The number of signature genes was determined by setting up significant changes of above two-fold (Table 1). The most remarkable finding of this analysis is that in the context of Tat, Meth sensitization had little if any impact on gene expression as compared with the controls. Moreover, the impact of Tat alone was very limited.

We used GeneVenn (http://genevenn.sourceforge.net/) to compare the genes that were members of the Tat−/Meth vs. Tat−/Sal and Tat+/Meth vs. Tat−/Meth signatures, both with a larger number of gene signatures, with trends subjected to quantile normalization. The degree of overlap between these two comparisons, Tat−/Meth vs. Tat−/Sal and Tat+/Meth vs. Tat−Meth, was 91% and 84%, respectively (Figure 2). Importantly, although the gene signatures resulting from these two comparisons showed substantial overlap, the interactions were heavily anticorrelated, indicating potential compensatory effects.

In the Tat+/Sal vs. Tat−/Sal comparison, the majority of the genes that were significantly changed overlapped with the other comparison signatures (nine out of 12 genes) (Figure 2). Fold change calculations for the highly differentially expressed genes for both comparisons verified that this overlap is correct. Moreover, the intensity values were similar across the replicates within the groups, indicating that the trends were not influenced by a single outlier.

The comparison between Tat+/Meth vs. Tat+/Sal did not show significant signatures.

We used the fold change calculations in individual comparisons, in upregulated genes, to make predictions of transcription factor usage patterns that could be useful for identifying potential interventional strategies and epigenetic regulators in future studies. These predictions were performed by applying the Match algorithm to the lists of signature genes, using TRANSFAC database versions 2018 and 2019.3 [58,59]. Transcription factor binding motif frequencies within the proximal promoter of upregulated genes (+500 to −200 bp) were calculated to distinguish groups based on potential usage, as previously described [29]. The lists of transcription factors that most likely get activated were generated based on the following two measures: (1) Matrix similarity (MSS), which is a score that describes the quality of a match between a transcriptional binding motif sequence expressed as a matrix, and arbitrary parts of the input gene list sequences, identifiable through the gene EnsemblID, and (2) the core similarity (CSS), which estimates the quality of the match in the five most conserved positions and used for ranking. Gene ontology, biological processes, molecular functions, and cellular component domains, as well as pathway enrichment analysis were performed in IPA using calculated Z-scores. Relevant gene networks were visualized using Genemania in Cytoscape, as described by us [29]. All specific comparisons were designed to identify the effects of Tat (Tat+/Sal vs. Tat−/Sal), the effect of Meth (Tat−/Meth vs. Tat−/Sal), the effect of Tat in the context of Meth (Tat+/Meth vs. Tat−Meth), and the effect of Meth in the context of Tat (Tat+/Meth vs. Tat+/Sal).

### 3.2. The Effects of Meth-Induced Sensitization: The Comparison between Tat−/Meth vs. Tat−/Sal Animals

The main effect of Meth was characterized by a downregulation of the majority of the genes, as revealed by the comparison between Tat−/Meth vs. Tat−/Sal groups, filtered by the FDR_BH-adjusted *p* value up to 0.05. These were 675 genes, functionally assigned to autoimmune disease (*p* = 0.002), natural killer cell-mediated cytotoxicity (*p* = 0.003), measles (*p* = 0.003), arachidonic acid metabolism (*p* = 0.003), cGMP dependent protein kinase G signaling pathway (*p* = 0.005), and phenylalananine metabolism (*p* = 0.009), indicating an effect on immune functions found to be decreased by Meth, based on the comparison between Tat−/Meth vs. Tat−/Sal groups. Table 2 shows the 40 most downregulated genes; among them, the neuronal cholinergic receptor alpha 2 (Chrna2) where both gain and loss of function and mutations have been described in association with multiple neurological disorders [61,62,63,64], with cannabis use [65], and with antisocial behaviors in adolescent drug users [66]. In this list of 50 genes, the strongest representation was of genes acting epigenetically in the nucleus (*p* = 0.0072) and transcription factor complexes (*p* = 0.001). A complete list of all signatures can be found in Appendix A.

There were only 37 genes that were significantly upregulated to above two-fold by Meth sensitization (Table 3). We focused on these genes to examine transcriptional factor usage triggered by this Meth sensitization model (Table 3). Among them, the serotonin (5-HT) receptor (HTR6), a G protein coupled receptor with high affinity for Meth, biomarker of mood disorders, and Meth-induced psychosis [67,68], and with relevance in Meth-associated behaviors [69,70] was 2.2-fold increased by Meth-induced sensitization (FDR_BH *p* = 0.02). These relevant signature genes that are shown in Table 3 were functionally assigned to endoplasmic reticulum (*p* = 0.009) and associated with diseases such as chemical dependence (*p* = 0.009), metabolic (*p* = 0.014), cardiovascular (*p* = 0.014), and neurological disorders (*p* = 0.01).

### 3.3. The Effects of Tat Expression: The Comparison Between Tat+/Sal vs. Tat−/Sal

In this model, Tat had a very modest effect on gene expression in the brain (Table 4). Of all the significantly changed genes, the cytochrome c oxidase unit 7b (Cox7B), which is essential for mitochondrial respiratory complex IV, was the only one that was more than two-fold cutoff and down modulated. Additionally, Tat had a strong effect in the context of Meth (described below).

### 3.4. The Effects of Tat Expression in the Context of Meth-Induced Sensitization: The Comparison Between Tat−/Meth and Tat+/Meth

There was a large overlap between genes that were significantly changed in Tat−/Meth and Tat+/Meth, although in different directions. In this comparison, HIV Tat appears to upregulate, or prevent the Meth-mediated downregulation, of most genes that were suppressed in the Tat−/Meth animals. Thus, in the Tat+/Meth group, the overall gene expression was more similar to the controls that were not treated with the drug and did not express Tat (Tat−/Sal, Figure 1). The complete list of signatures, thresholds and *p* values can be found in Appendix A.

### 3.5. The Effects of Meth-Induced Sensitization in the Context of Tat Expression: The Comparison Between Tat+/Meth versus Tat+/Sal

Using FDR_BH adjusted *p* value alpha 0.05, the effect of Meth in the context of Tat did not result in above two-fold significant signatures.

### 3.6. Prediction of Transcriptional Signatures

The analysis of transcription factor binding motif frequencies in the promoters of genes upregulated by Meth sensitization (Table 3) led to the prediction of potential regulators activated by Meth, with potential value in early disease and as signatures. Transcriptional factor motif frequency predictions were also performed for effects of Tat in the Meth context. Table 5 shows the most frequent transcription factor binding motifs that were identified in predictions derived from the comparison between Tat−/Meth versus Tat−/Sal brains, as well as of genes upregulated by Tat in the context of Meth, derived from the comparison between Tat+/Meth versus Tat−/Meth brains (Section 3.4). We found that 26% of the genes upregulated in Tat+/Meth as compared with Tat−/Meth, were genes that contained one or two TATA box sequence domains in their proximal promoters (Table 5). This finding is in agreement with our previous findings in vitro, of the enrichment of TATA-box binding protein (TBP) and of genes bearing this motif by Tat upon interaction with Meth [29]. In our previous studies, we found that the TATA box-bearing genes that are affected by the interaction between Tat and Meth are mostly inducible early response genes, particularly involved in stress responses and the immune system [29]. Yet, it is noticeable that in the context of Meth sensitization, the impact of Tat almost exclusively resulted in the reversal of Meth effects, returning the expression of Meth-altered genes to resemble Tat−/Sal levels. In the absence of Tat, TATA-box usage was not found to be significant in TRANSFAC simulations (Table 5).

A reverse analysis of transcription factor binding motif frequency in all conditions, and all perturbed genes, was performed under the hypothesis that our predictions could be further tested, and that major targets could be potentially identified. For that, we used iRegulon to generate nodes containing the targets of each one of the transcription factors in Table 5, identified in the data, with a focus on the ones that were significantly associated with both effects of Meth, and effects of Tat in the context of Meth. The generated clusters were merged, creating a network with the following two main components: The largest component integrated target genes for Kruppel-like factor 6 (KLF6), myogenin (MYOD1), AP-1 with Fos and Jun interactions, tumor protein 53 (TP53), and the cAMP response element-binding protein 1 (CREBP1) and a smaller component had Zink finger 333 (ZNF333) as a central transcription factor (Figure 3). The analysis of this two-component network showed a clustering coefficient of 0.024, and 2.59 average neighbors, suggesting interconnections through intermediate nodes (labeled with a red border). For instance, serine-threonine protein kinase AKT1 and the centromere protein F (CENPF) are regulated by both KLF6 and MYOD1, Bcl-2 associated X (BAX) is regulated by both MYOD1 and TP53, and the vimentin gene (VIM) is regulated by both KLF6 and AP-1 (JUN/FOS). Moreover, ephrin receptor A2 (EPHA2), growth arrest and DNA-damage-inducible alpha (GADD45A), stratifin (SFN), Fos-related antigens 1 (FOSL1), and B (FOSB) are regulated by both AP-1 and TP53, while the genes of Maf-bZip transcription factor (MAFF), activating transcription factor 3 (ATF3), tyrosine 3-monooxygenase/tryptophan 5-monooxygenase activation protein zeta (YWHAZ), and the dual specific protein phosphatase 1 (DUSP1) are in the interface between AP-1 and CREBP1. The gene of cyclin-dependent kinase inhibitor 1A (CDKN1A) was predicted to be regulated by all the predicted transcription factors. Interestingly, ZFN333 segregated into a smaller component without connectivity with the larger one.

When comparison input data was introduced in this interconnected transcription factor regulatory network of genes, some patterns could be visualized (Figure 3). For instance, compared to Tat−/Sal controls, both Tat alone (Tat/Sal) (Figure 3A), and Meth alone (Tat−/Meth) (Figure 3B), had an effect on MYOD1 and its regulated genes in the interface with KLF6 regulation. Yet, KLF6 itself and other immediate neighbors were not significantly disturbed by these conditions (Figure 3A,B). Targets of TP53, such as interleukin 1b (IL1b), are significantly increased by Tat alone and by Meth alone. Homeobox proteins that are targets of ZNF333, are two- to four-fold increased by Tat or Meth, but these changes were not significant. When Tat was expressed in Meth challenged animals (Tat+/Meth), a stronger effect was noticed on the upregulation of AP1 complex genes, and their targets, with an effect on intermediate effectors between AP-1 and CREBP1, ATF3, and FOSB by Meth in the context of Tat (Figure 3C), and DUSP1, and FOSB by Tat in the context of Meth (Figure 3D). The TP53 target TNFSF103 was significantly but below three-fold elevated in Tat+/Meth animals, as well as GADD45A, which was also regulated by AP-1 (Figure 3D). This systems approach has generated a transcriptional regulation hypothesis, where the overall effect of Tat in the context of Meth can be the activation of AP-1 as a major contributor that modifies the effects of Meth alone, as well as Tat alone.

### 3.7. An Unfastened Analytical Strategy Identifies Overlapping Pathways Significantly Affected by Meth, and by Tat+/Meth, in Anti-Directional Ways

Our analysis has shown a main effect of Meth-induced sensitization and an anti-directional effect of Tat in the context of Meth. Therefore, we focused on these interactions. For this investigation of interactions, the data was filtered with calculated z-scores that identified matches between expected and observed relationship directions between genes, with a cutoff raw *p*-value alpha 0.01, and the prediction of gene networks and pathways where changes in the model suggest strong action.

The top five overlapping canonical pathways and the *p* values for each designated comparison were mitochondrial dysfunction (Figure 4, Tat−/Meth vs. Tat−/Sal *p* = 4.80 × 10^−15^ and Tat+/Meth vs. Tat−/Meth *p* = 2.19 × 10^−11^, with 65.5% and 57.9% overlap, respectively), oxidative phosphorylation (Figure 5, Tat−/Meth vs. Tat−/Sal *p* = 1.56 × 10^−14^ and Tat+/Meth vs. Tat−/Meth *p* = 4.50 × 10^−12^, with 73.9% and 67.0% overlap, respectively), EIF2 signaling (Figure 6, Tat−/Meth vs. Tat−/Sal *p* = 7.37 × 10^−12^ and Tat+/Meth vs. Tat−/Meth *p* = 6.03 × 10^−13^, with 58% and 56.4% overlap, respectively), sirtuin signaling pathway (Figure 7, Tat−/Meth vs. Tat−/Sal *p* = 4.03 × 10^−8^ and Tat+/Meth vs. Tat−/Meth *p* = 3.24 × 10−7, with 50.4% and 46.4% overlap, respectively) and regulation of eIF4 and p70S6K signaling (Figure 8, Tat−/Meth vs. Tat−/Sal *p* = 7.55 × 10^−6^ and Tat+/Meth vs. Tat−/Meth *p* = 7.53 × 10^−6^, with 51.8% and 48.9% overlap, respectively). A complete list of all the genes in each one of these pathways, thresholds and *p* values can be found in Appendix A. Upstream regulators identified in the model included a highly significant activation of the rapamacyn-insensitive companion of mTOR (RICTOR) by Meth exposure (*p* = 1.68 × 10^−15^), and disorders assigned to cancer (*p* = 3.2 × 10^−11^), organismal injuries (*p* = 3.2 × 10^−11^), endocrine system disorders (*p* = 8.83 × 10^−11^), and metabolic disease (*p* = 4.3 × 10^−9^).

The pathways most significantly affected by the interactions, with overlapping actions observed in Tat−/Meth vs. Tat−/Sal and in Tat+/Meth vs. Tat−Meth, were predominantly characterized by suppression of gene expression by Meth-induced sensitization and a counteracting positive effect of Tat expression. These pathways had recurrent and redundant gene representations heavily assigned to mitochondrial functions, which can be visualized in Figure 4, Figure 5, Figure 6, Figure 7 and Figure 8. The sixth top canonical pathway is a function of dopamine (*p* = 9.76 × 10^−5^ for Tat−/Meth vs. Tat−/Sal, and *p* = 2.45 × 10^−5^ for Tat+/Meth vs. Tat−/Meth), which we have previously described [43].

A large number of NADH ubiquinone oxidoreductase supernumerary subunit (NDUF) genes were represented in the two most significant pathways with anti-directional disturbances in Tat−/Meth vs. Tat−/Sal and in Tat+/Meth vs. Tat−/Meth, which were the mitochondrial dysfunction pathway (Figure 3) and in the oxidative phosphorylation pathway (Figure 5). Many of these were also represented in the fourth most significant sirtuin signaling pathway (Figure 7). Together, the effect of Meth-induced sensitization on suppressing these pathways suggests a strong impact of the drug in energy production, homeostasis, and balance, as well as a major effect on mitochondrial health. Interestingly, the interaction of Meth sensitization with Tat expression shows that the Tat protein, when expressed in the absence of a productive infection, can elicit a protective response, with a few exceptions. For instance, the cytochrome c oxidase subunit 7A2 (Cox7A2l) and the somatic cytochrome C (Cycs), which were both suppressed by Meth, were not recovered by Tat expression, as seen in the Tat+/Meth vs. Tat−/Meth comparison, where these genes are shown in green, indicating downregulation (Figure 4B,D).

The third most significantly affected pathway, by Meth-induced sensitization and by the interaction between Meth exposure and Tat, was the eukaryotic initiation factor 2 (EIF2) pathway (Figure 6) that is a stress-driven pathway response with a primary role regulating mRNA translation and protein synthesis [71], which further suggests that Meth exposure is a potential suppressor input and HIV Tat is an activator. Several genes in this pathway were also represented in the regulation of eIF4 and p70S6K signaling (Figure 8).

The sirtuin signaling pathway was the fourth most significantly affected pathway (Figure 7). The Meth-induced sensitization regimen caused a significant transcriptional decrease of all sirtuins and of several components that are regulated by sirtuins, including those associated with inflammation, such as the inducible nitric oxide synthase (NOS2), the intercellular adhesion molecule 1 (ICAM1), and with aging, such as the β-amyloid precursor protein (β-APP). With a few exceptions in the sirtuin pathway, Tat expression also had a predominantly preventative effect upon the interaction with Meth. The molecules in which expression was not recovered by Tat, in the context of Meth sensitization, are shown in green indicating downregulation (Figure 7B). They included APP, the mitochondrial marker and import receptor TOMM22, the ATP citrate lyase (ACLY), which is involved in the synthesis of acetylcholine, the RNA polymerase subunit 1A (PolR1A), and the epigenetic silencer sirtuin 1 (SIRT1). The downregulation of APP and of SIRT1 in the model were further and successfully validated by qRT-PCR (Figure 9A,B). Other genes selected for validation include were decreased by Meth and recovered in the context of Tat expression. These include NOS2 (Figure 9C) and PPARa (Figure 9D).

The genes involved in the regulation of eIF4 and p70S6K/mTOR pathway play a regulatory role in mRNA transcription and translation [72] (Figure 8). The Meth–induced sensitization regimen suppressed the expression of the genes in this pathway with an anti-directional effect of Tat interaction.

### 3.8. Validation of Genes of Interest

We used qRT-PCR to validate strong gene signatures of interest that were significantly downregulated both by Tat expression and by Meth exposure, independently and together, and that could have consequences that replicate observations in human HIV neurological disorders, as per our previous observations in other models of neuroHIV. For instance, by qPCR, the transcriptional levels of SIRT1 were significantly decreased by Tat induction, as well as by Meth exposure, and also by their interaction (Figure 9A), replicating what we previously reported in the SIV rhesus macaque model and in correlation with a potentially accelerated aging phenotype [73,74]. We also validated the decrease in APP caused by Tat and by Meth alone (Figure 9B). The interaction between Tat and Meth caused a partial recovery in the transcription of APP as compared with Meth alone, but it was not sufficient to reach control levels (Figure 9B). Other genes selected for validation included those that were decreased by Meth and recovered in the context of Tat expression. These included NOS2 (Figure 9C), and PPARa (Figure 9D).

## 4. Discussion

We examined the hypothesis that HIV Tat expression and Meth independently modify gene transcription in the brain during sensitization, and have interactive effects that result in patterns associated with disease and with triggered locomotor behaviors, which have been previously described in the same animals [43]. The effect of Tat was determined in males, given the confounders of the estrous cycle that are commonly observed in female mice. We applied conservative analytical strategies to reduce false positives, using raw data filtered at three-fold above background, two-fold changes upon group comparisons, and with an FDR alpha 0.05, and a raw *p* value of 0.01. This approach was chosen based on robust and consistent validation of candidate genes as compared with lower cutoff settings (not shown).

The analysis of the transcriptional profile generated by Tat expression in the brain of transgenic mice as compared with controls, on the one hand, revealed a limited impact associated with the absence of strong gene signatures. The Meth-induced sensitization, on the other hand, had a very severe impact on gene expression with a strong focus on pathways within mitochondria and endoplasmic reticulum. The suppression of mitochondrial function by Meth has been extensively described in the brain and in tissues that are rich in these organelles [75,76,77,78]. Meth-associated stress to the endoplasmic reticulum has been associated with blood–brain barrier disruption [79], and loss of astrocytes by apoptosis [80]. In addition, an attenuation of inflammatory response by Meth has been previously observed [81], by preventing the neuro-immune system signal in response to systemic infectious stimuli such as bacterial lipopolysaccharide. The impact of the Meth exposure on enhancing and exacerbating inflammation has been shown to be higher in aged animals [82] or with higher, neurotoxic doses of the drug [83].

Our study was limited to the transcriptome, and more significant changes in the presence of Tat could potentially be observed at the proteome level, which was not examined here. A potential leaky Tat mRNA expression could also be responsible for diluting strong signatures in the model [18] Nevertheless, the mouse model tested here allows a rigorous systems-based analysis of the effects of upregulation of the Tat protein, although excluded from active infection, and proposes a model on how it can act in vivo on modulating suppressive patterns of transcription changes that are caused by comorbidities such as Meth. The possibility that Tat is able to modulate cis-negative repressor elements that are induced by Meth, remains to be examined. It is known that Tat functions through master transcriptional regulators bound at promoters and enhancers, rather than through cellular”TAR-like” motifs, to both activate and repress host gene sets sharing common functional annotations [84]. We have previously shown that TATA-box binding peptide is critical for generating inflammatory signatures in vitro [29]. However, we did not identify strong inflammatory signatures in this transgenic mouse model. Instead, our systems biology-based analysis suggests that Tat is able to restore suppressive signals by Meth through effects on mitochondria and transcription initiation complexes.

The prediction of upstream regulators that are involved in Meth exposure led to a potential role of RICTOR, a rapamycin-insensitive component of the mTOR complex with a role in cell cycle [85], but also described in association with dopamine-dependent behaviors and deficits in dopaminergic areas of the brain [86]. Importantly, we previously tested these animals for the effects of Meth on the dopaminergic system, however, Tat expression enhanced motor sensitization and did not restore dopaminergic changes caused by Meth [43]. This highlights a limitation of this study which is the dissociations between behavioral phenotypes and gene expression signatures, but also demonstrates the value of this experimental animal model.

The predictions of promoter target motif were highly informative. Meth alone or in the context of Tat expression showed consistency of target motifs in genomic promoters, including binding motifs for KLF6, ZNF333, AP-1, CREBP1, p53 (TP53), and myogenin (MYOD1). By modeling the interactome of targets of these transcription factors, common effectors could be visualized and suggest redundancies that are not easily dissected using systems biology approaches but need further examination. For instance, on the one hand, KFL6 and MYOD1 can be involved in activating common targets when Tat alone or Meth alone are compared to controls. Yet, these two conditions cause a transcriptional increase in MYOD1, but not KLF6. The transcriptional activity of KLF6 cannot necessarily be associated with an enhanced transcription. In addition, targets that are not common between these two transcription factors exhibit a low activation. On the other hand, Tat+/Meth animals show an enhanced FOS transcription, and also enhanced activity on AP-1 exclusive targets, as well as targets that are common to TP53 and CREBP1. These hypothesis-generating modeling methodologies can test the weight of potential transcriptional regulators, but they need to be further tested in cell systems, to overcome the limitations of the systems biology approach.

Most importantly, in the context of Meth exposure, the HIV Tat protein counteracted the effects of Meth on gene transcription, driving the global expression pattern to more closely resemble what is observed in non-sensitized Tat-negative saline-treated (Tat−/Sal) controls. Therefore, in this transgenic Tat mouse model, Tat expression in the brain in the absence of Meth sensitization could not have strongly modified transcriptional profiles to the levels observed in active infection, or in HIV associated neurological disorders (HAND) in humans [87,88,89,90,91,92]. Nevertheless, it had an anti-directional and potentially positive impact in the context of interaction with Meth exposure. The results found, in this study, for effects of Tat expression differ from what has been described regarding, for example, inflammatory genes, both in vitro and in vivo [29,93,94], and including this mouse model [18,33]. Gliosis and neurological changes that have been reported in Tat transgenic animals, however, could result from post-transcriptional effects in models that do not involve drug abuse interactions [33]. Hypothetically, Meth sensitization could induce cis-regulatory repression elements, in addition to decreasing the expression of translation initiation complexes, such as the eIF2, eIF4, and p70S6K signaling pathways, which were identified here as presenting significant disruptions in Meth. These pathways control translation and transcription, especially of mRNAs that present a secondary structure [95,96], with implications for the CNS. For example, deficits and mutations in eIF2 and eIF4 pathways have been associated with neurological disorders and aging [96,97]. The fact that these pathways are deeply affected by Meth, and their recovery in the presence of Tat, suggests the potential involvement of upstream mTOR [98,99,100,101,102] as a critical element in Tat along with Meth sensitization, and as a response to mitochondrial oxidative stress [103,104]. In the presence of Tat, mitochondrial dysfunction and oxidative phosphorylation pathways were also maintained at control levels. Moreover, Meth and Tat enhanced the role of AP-1 complexes, which is responsible for a range of effects, including contributing to neutralizing *cis*-negative regulatory elements, a function that is deeply associated with promoting viral transcription [105], but potentially exerting effects on bystander host transcriptional patterns that are repressed by Meth. The role of ROS to these effects must be further clarified. A summary of the hypothesis generated by our analysis is shown in Figure 10.

An interesting finding is that the induction of Tat expression in the brain did not prevent the decrease in SIRT1 transcription that was caused by Meth sensitization. We have shown that SIRT1 transcription is suppressed by active infection [73,74] and that this decrease is a key factor in the development of long-term disruptions in the regulation of gene expression, with resulting aging-like phenotypes. Likewise, APP transcription was suppressed by Meth sensitization, by Tat expression, as well as by their interaction. Although a decrease in APP is regarded as protective, it could be a factor limiting β-APP supply [106,107,108]. Whether these are replicating human disease and showing signs of accelerated aging in this model, as a consequence of Meth and HIV Tat interaction, needs to be further examined.

Regarding the interactions of Tat and Meth, our findings were surprising. As mentioned, our previous work has shown that these Meth sensitized animals experience an important repression in the dopaminergic system that is further enhanced by Tat [43], resulting in higher locomotor responses to the drug challenge and replicating aspects of the pathology found in HIV-infected humans [43,109,110]. In chronic and binge Meth administration regimens performed in the same mouse model, the induction of Tat expression during the final cycle of Meth exposure did not impact brain reward function during withdrawal. Nevertheless, there was a trend that Tat expression could contribute to an increased severity of withdrawal in the binge regimen as compared with the chronic regimen [109]. Overall, this finding suggests a subtle effect of Tat expression on brain reward function during Meth withdrawal [109]. Thus, while the effects of Tat alone were subtle, the Tat transgenic mouse model is a tool to examine the role of the HIV Tat protein in influencing the response to Meth in the different paradigms used to study addiction.

Our results have shown that in this transgenic mouse model, Tat in the brain, in the absence of a replicating virus, did not induce strong gene signatures at challenge. However, the effects of Tat were not absent, since in the context of interaction with Meth sensitization and exposure, it caused transcription patterns to become overall more similar to controls at challenge, counteracting the effects of Meth. It is possible that the ability of Tat to quickly translocate and epigenetically trigger transcription in host cells [29] could serve as a compensatory mechanism to counteract the suppressing effects of Meth exposure, by restoring the levels of initiation complexes. However, in the absence of the factors that are present during active replication, the Tat protein possibly is not sufficient to exacerbate gene transcription.

It is important to acknowledge that this study is limited to transcriptional changes and post transcriptional modifications that can affect phenotypes were not examined here.

In conclusion, we have shown that the HIV Tat transgenic mouse is a tool for the examination of in vivo interactions between the HIV Tat protein and exposure to drugs of abuse. The effects of Tat in this transgenic mouse model are found to be limited and focal when stringent statistical measures are taken. Meth sensitization has a strong effect on pathways associated with mitochondrial health and protein translation, and the induction of Tat in that context positively reverted the effects of Meth sensitization, which were detectable after a withdrawal period and at challenge. The effects of Meth and the countering effects of Tat are largely associated with transcriptional regulation mechanisms that can combine silencing and activation of transcriptional factors with binding motifs in redundant target promoters (Figure 10).

While the overall observations were not in agreement with our hypothesis, they demonstrate the challenges and the values of animal models in replicating aspects of human HIV neurological consequences under rigorous systematic conditions, especially in the context of drug comorbidities and different drug administration and sensitization paradigms.

## Figures and Tables

**Figure 1 viruses-12-00426-f001:**
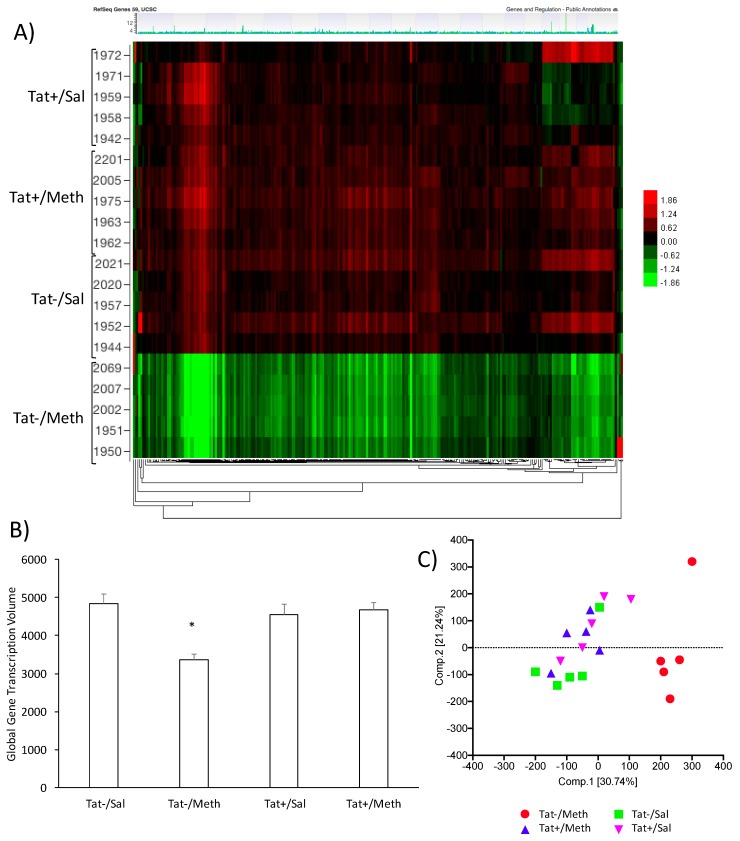
The effects of Methamphetamine (Meth) and trans-activator of transcription (Tat) expression on transcriptional activity in the brain. (**A**) Heat map clustering of gene signatures generated between Tat− and Tat+ mice sensitized with Meth or saline (Sal) prior to the Meth challenge. The caudate-putamen tissue was evaluated for gene expression using an Agilent mouse gene array platform. The heat map shows patterns of expression normalized to the average expression of all the data. Horizontal rows represent individual animals, clustered in groups, Tat+/Sal, Tat+/Meth, Tat−/Sal, and Tat−/Meth. The numbers refer to each individual mouse identification. Vertically, genes are arranged in clusters according to change patterns between groups. Expression values are represented in colors according to the levels of upregulation (red shades), downregulation (green shades), or no change (black); (**B**) Global transcriptional volume calculated using the sum of all up- and downregulated fold changes group averaged for each group condition; (**C**) Principal component analysis (PCA) scores for individual animals. PCA was performed using Tat expression and sensitization for quality control and to summarize the ways in which gene responses vary under Tat, Meth, or their interaction.

**Figure 2 viruses-12-00426-f002:**
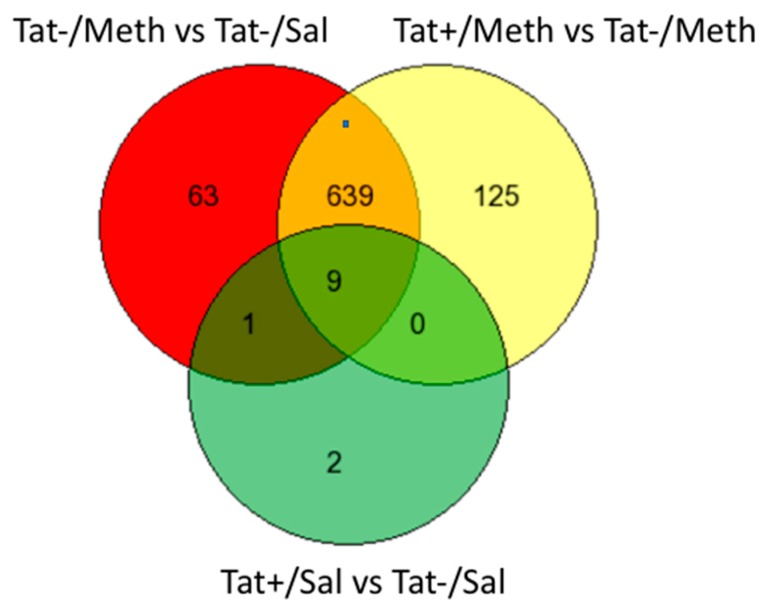
Venn diagram showing the number of exclusive and overlapping signatures in assigned comparisons, regardless of direction, in different assigned comparisons. The comparisons between groups were established in order to identify the effect of Tat (Tat+/Sal vs. Tat−/Sal), the effect of Meth (Tat−/Meth vs. Tat−/Sal), the effect of Tat in the context of Meth (Tat+/Meth vs. Tat−Meth), and the effect of Meth in the context of Tat (Tat+/Meth vs. Tat+/Sal). There were 712 gene signatures identified for the comparison of Tat−/Meth with Tat−/Sal, of which 63 were exclusive to the effect of Meth. Tat+/Meth as compared with Tat−/Meth produced 773 gene signatures of which 125 were exclusive to the effect of Tat in the context of Meth. Between these two comparisons, a total of 648 genes overlapped in an anti-directional manner. The comparison, Tat+/Sal vs. Tat−/Sal, identified 12 gene signatures of which only two were exclusive to the effects of Tat. Nine of all signatures overlapped between all three comparisons. Since the comparison between Tat+/Meth and Tat+/Sal did not produce significant signatures, it is not included in this graph.

**Figure 3 viruses-12-00426-f003:**
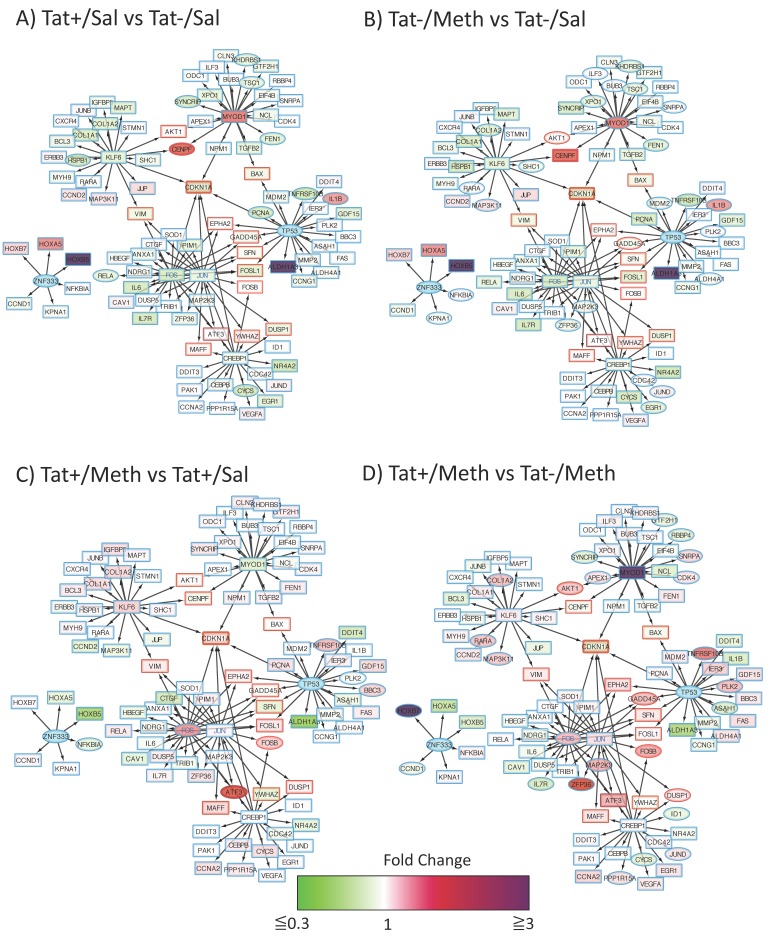
Modeling of the interactions between predicted dominant transcription factors and their targets. Genes that were significantly modified by Meth alone and by Tat in the context of Meth were searched using iRegulon in Cytoscape, to identify targets of commonly relevant transcription factors according to TRANSFAC simulations. Then, individual networks were merged for the identification of common targets. Merged components were used for visualization of the impact of individual conditions by applying fold change data attributes. Red/magenta nodes represent upregulated and green nodes are downregulated changes. Elliptical shapes represent statistically significant changes. Red lines represent common targets between two or more transcription factors. Blue nodes represent transcription factors found to be relevant in TRANSFAC simulations, but not represented in the data. Each coded comparison allows the visualization of (**A**) the effects of Tat from the comparison of Tat+/Sal vs. Tat−/Sal; (**B**) the effects of Meth from the comparison of Tat−/Meth vs. Tat−/Sal; (**C**) the effects of Meth in the context of Tat, from the comparison between Tat+/Meth vs. Tat+/Sal; and (**D**) the effects of Tat in the context of Meth from the comparison between Tat+/Meth vs. Tat−/Meth.

**Figure 4 viruses-12-00426-f004:**
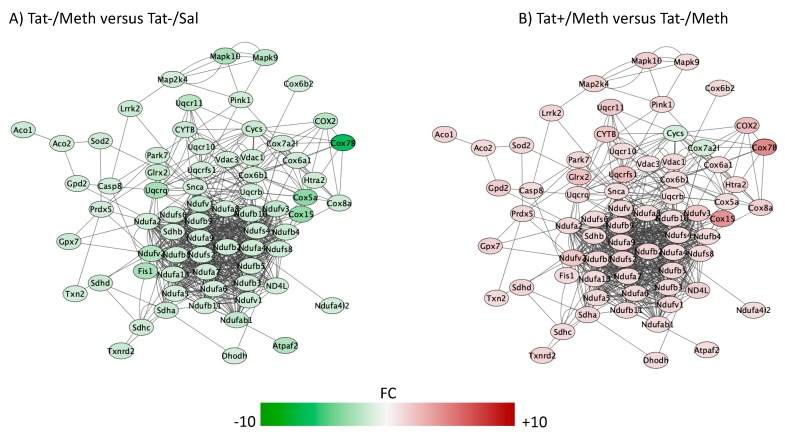
Mitochondrial dysfunction pathway. Genes with overlapping actions observed in (**A**) Tat−/Meth vs. Tat−/Sal and in (**B**) Tat+/Meth vs. Tat−/Meth and assigned to mitochondrial dysfunction were loaded into Cytoscape via Genemania; significant changes (*p* < 0.01) associated to assigned comparisons were visualized by loading node attributes. Tat−/Meth vs. Tat−/Sal *p* = 4.80 × 10^−15^ and Tat+/Meth vs. Tat−/Meth *p* = 2.19 × 10^−11^, with 65.5% and 57.9% overlap, respectively.

**Figure 5 viruses-12-00426-f005:**
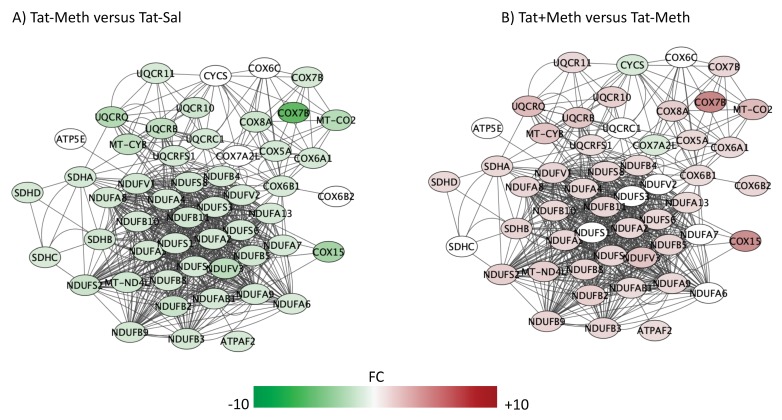
Oxidative phosphorylation pathway. Genes with overlapping actions observed in (**A**) Tat−/Meth vs. Tat−/Sal and in (**B**) Tat+/Meth vs. Tat−/Meth and assigned to oxidative phosphorylation were loaded into Cytoscape via Genemania; significant changes associated to assigned comparisons (*p* < 0.01) were visualized by loading the node attributes table. Tat−/Meth vs. Tat−/Sal *p* = 1.56 × 10^−14^ and Tat+/Meth vs. Tat−/Meth *p* = 4.50 × 10^−12^, with 73.9% and 67.0% overlap, respectively.

**Figure 6 viruses-12-00426-f006:**
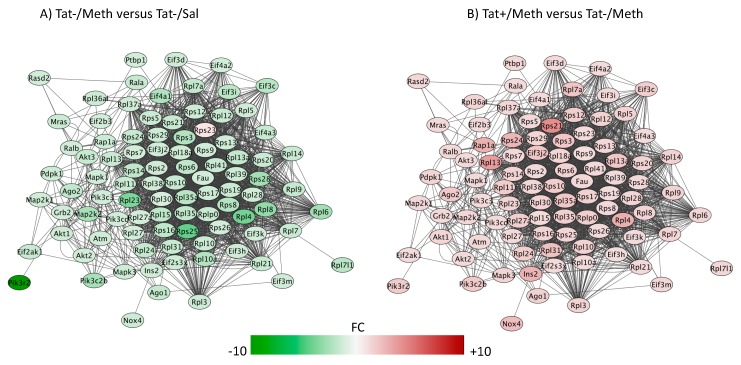
Eukaryotic initiation factor 2 (EIF2) signaling pathway. Genes with overlapping actions observed in (**A**) Tat−/Meth vs. Tat−/Sal, and in (**B**) Tat+/Meth vs. Tat−/Meth and assigned to EIF2 signaling were loaded into Cytoscape via Genemania; significant changes associated to assigned comparisons (*p* < 0.01) were visualized by loading the node attributes table. Tat−/Meth vs. Tat−/Sal *p* = 7.37 × 10^−12^ and Tat+/Meth vs. Tat−/Meth *p* = 6.03 × 10^−13^, with 58% and 56.4% overlap, respectively.

**Figure 7 viruses-12-00426-f007:**
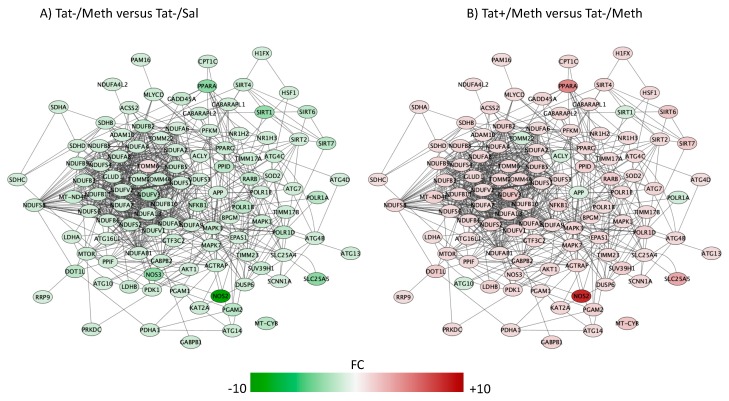
Sirtuin signaling pathway. Genes with overlapping actions observed in (**A**) Tat−/Meth vs. Tat−/Sal and in (**B**) Tat+/Meth vs. Tat−/Meth and assigned to sirtuin signaling were loaded into Cytoscape via Genemania; significant changes associated to assigned comparisons (*p* < 0.01) were visualized by loading the node attributes table. Tat−/Meth vs. Tat−/Sal *p* = 4.03 × 10^−8^ and Tat+/Meth vs. Tat−/Meth *p* = 3.24 × 10^−7^, with 50.4% and 46.4% overlap, respectively.

**Figure 8 viruses-12-00426-f008:**
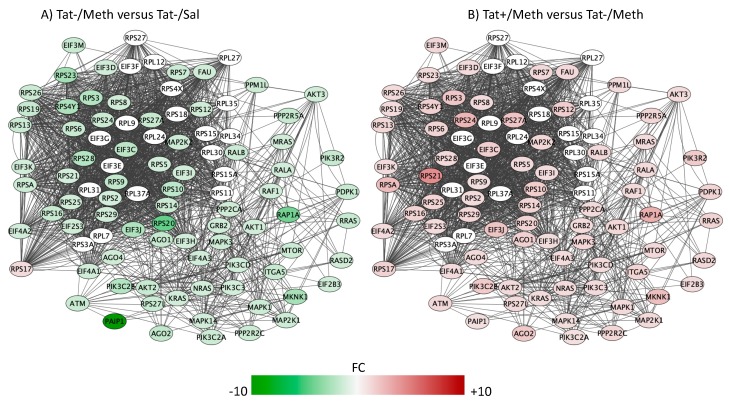
Regulation of the eukaryotic initiation factor 4 (eIF4) and p70S6K signaling. Genes with overlapping actions observed in (**A**) Tat−/Meth vs. Tat−/Sal and in (**B**) Tat+/Meth vs. Tat−/Meth, and assigned to the regulation of eIF4 and p70S6K signaling were loaded into Cytoscape via Genemania; significant changes associated to assigned comparisons (*p* < 0.01) were visualized by loading the node attributes table. Tat−/Meth vs. Tat−/Sal *p* = 7.55 × 10^−6^ and Tat+/Meth vs. Tat−/Meth *p* = 7.53 × 10^−6^, with 51.8% and 48.9% overlap, respectively.

**Figure 9 viruses-12-00426-f009:**
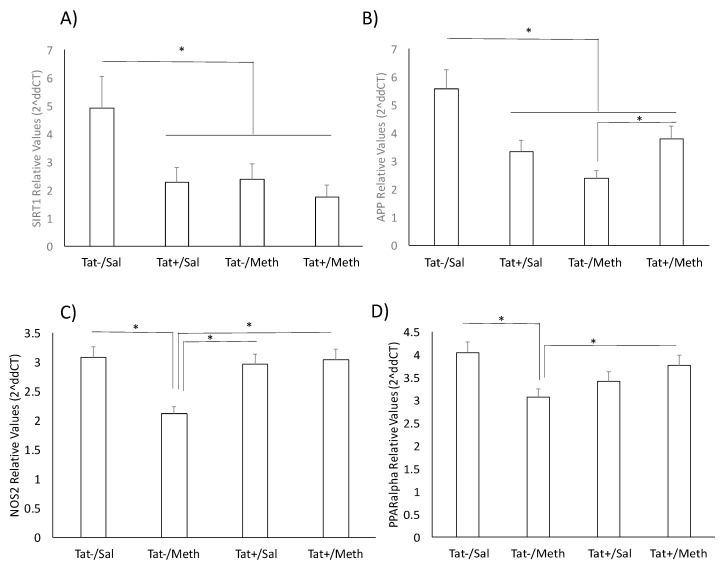
PCR validation of key genes not affected by the interaction between Tat expression and Meth during sensitization. The primers for (**A**) SIRT1; (**B**) amyloid precursor protein (APP); (**C**) NOS2; and (**D**) PPARa were purchased from Qiagen, and the expression was normalized by the average of 2 housekeeping genes, 18S and GAPDH. * *p* < 0.05 in ANOVA followed by Bonferroni’s post hoc test between assigned comparisons.

**Figure 10 viruses-12-00426-f010:**
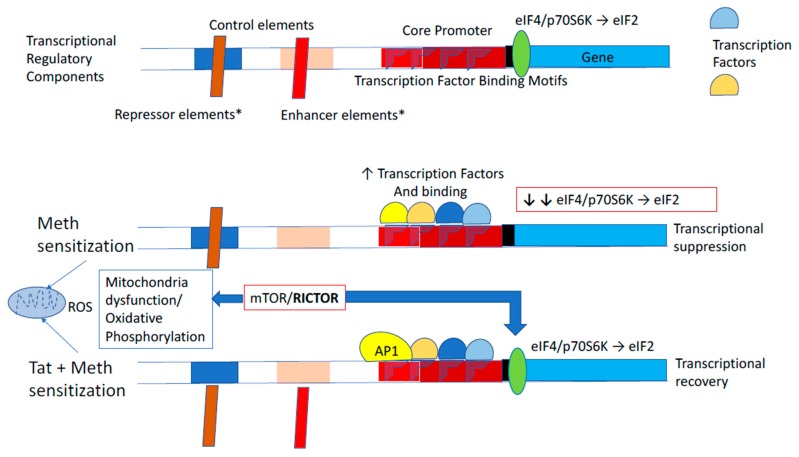
Summary figure of the systems biology-generated hypothesis. Studies on transcriptional patterns, pathway analysis, and transcription factor usage predictions in the brain of Tat-transgenic mice sensitized with Meth suggest the following: Both Meth sensitization alone, and along with the induction of HIV Tat protein, activate transcriptional regulatory mechanisms affecting the same genes but in opposite ways. Overlapping transcriptional factor usage between these two conditions are predicted and could be further regulated by differential actions on transcriptional control elements outside the core promoter, as well as levels of transcription and translation initiation factors (eIF2 and eIF4). These differences can be orchestrated by pathways in the mitochondria, including production of reactive oxygen species (ROS) contributing to changes in oxidative phosphorylation, with mTOR/RICTOR complex as a central regulator, and with AP1 support. We disclose that this concept was generated by a systems biology analytical strategy, and further experiments are necessary.

**Table 1 viruses-12-00426-t001:** Number of signatures derived from group comparisons for gene expression in brains from Tat+ and Tat− mice treated with Meth or with saline, with FDR_BH < 0.05, and absolute fold change >2.0 upon assigned comparisons.

Description	Number of Genes	% of Genes	Number Downregulated	Number Upregulated
Tat−/Meth vs. Tat−/Sal	712	3.85	675	37
Tat+/Meth vs. Tat+/Sal	0	0	0	0
Tat+/Meth vs. Tat−/Meth	773	4.18	44	729
Tat+/Sal vs. Tat−/Sal	1	0.005	1	0

**Table 2 viruses-12-00426-t002:** Forty most downregulated genes in Tat−/Meth animals as compared with Tat−/Sal.

GeneSymbol	Gene Name	Tat−/Meth vs. Tat−/Sal Fold Change	FDR_BH *p* Value
Bard1	BRCA1 associated RING domain 1	−50.62054	1.49 × 10^−12^
Sall4	sal-like 4 (Drosophila)	−36.81502	1.31 × 10^−11^
Krtap10-10	keratin associated protein 10-10	−28.0749	9.03 × 10^−15^
Krtap8-2	keratin associated protein 8-2	−27.74784	5.26 × 10^−11^
Olfr1350	olfactory receptor 1350	−26.34379	7.71 × 10^−8^
Klk11	kallikrein related-peptidase 11	−25.88588	6.23 × 10^−7^
Krtap5-5	keratin associated protein 5-5	−25.47586	3.83 × 10^−12^
Sox13	SRY-box containing gene 13	−25.23627	6.90 × 10^−13^
Elane	elastase, neutrophil expressed	−23.62923	1.70 × 10^−13^
Baiap2l2	BAI1-associated protein 2-like 2	−22.7781	3.03 × 10^−14^
Obp1a	odorant binding protein Ia	−22.46356	7.77 × 10^−11^
C87414	expressed sequence C87414	−21.31501	4.89 × 10^−2^
Chrna2	cholinergic receptor, nicotinic, alpha polypeptide 2 (neuronal)	−20.67663	7.91 × 10^−12^
Vax2	ventral anterior homeobox containing gene 2	−20.52762	3.65 × 10^−13^
Myod1	myogenic differentiation 1	−19.55757	3.01 × 10^−6^
Lman1l	lectin, mannose-binding 1 like	−19.06574	8.39 × 10^−8^
Tbx10	T-box 10	−19.04252	3.88 × 10^−7^
Uimc1	ubiquitin interaction motif containing 1	−18.35725	1.63 × 10^−9^
Adat1	adenosine deaminase, tRNA-specific 1	−17.68073	7.22 × 10^−11^
Notum	notum pectinacetylesterase homolog (Drosophila)	−17.47123	8.13 × 10^−14^
4Sun5	Sad1 and UNC84 domain containing 5	−17.39883	5.34 × 10^−8^
H2−Ob	histocompatibility 2, O region beta locus	−16.88096	5.41 × 10^−14^
Msx2	homeobox, msh-like 2	−16.50882	7.78 × 10^−7^
Pgc	progastricsin (pepsinogen C)	−15.78614	1.38 × 10^−11^
Slc45a3	solute carrier family 45, member 3	−15.57587	3.57 × 10^−13^
Aldh3a1	aldehyde dehydrogenase family 3, subfamily A1	−15.56452	1.23 × 10^−13^
Apobec3	apolipoprotein B mRNA editing enzyme, catalytic polypeptide 3	−15.04451	2.72 × 10^−14^
Cd79a	CD79A antigen (immunoglobulin-associated alpha)	−14.53746	1.52 × 10^−6^
Zglp1	zinc finger, GATA-like protein 1	−14.42331	1.60 × 10^−13^
Tm4sf5	transmembrane 4 superfamily member 5	−14.26034	6.67 × 10^−11^
Pom121l2	POM121 membrane glycoprotein-like 2 (rat)	−14.03769	2.06 × 10^−11^
Gng4	guanine nucleotide binding protein (G protein), gamma 4	−13.83425	6.72 × 10^−7^
Cym	chymosin	−13.65853	9.79 × 10^−14^
Tsga10ip	testis specific 10 interacting protein	−13.288	1.21 × 10^−9^
Msi1	Musashi homolog 1(Drosophila)	−13.23837	7.31 × 10^−11^
Olfr1243	olfactory receptor 1243	−12.73216	1.03 × 10^−8^
Atp6v1g3	ATPase, H+ transporting, lysosomal V1 subunit G3	−11.99044	4.40 × 10^−10^
Casp14	caspase 14	−11.97585	2.44 × 10^−12^
Tgif2lx1	TGFB−induced factor homeobox 2-like, X-linked 1	−11.71554	8.92 × 10^−8^
Tmem146	transmembrane protein 146	−11.54644	1.41 × 10^−10^

**Table 3 viruses-12-00426-t003:** Genes significantly affected by Tat−/Meth as compared with the Tat−/Sal group.

Gene Symbol	Gene Name	Tat−/Meth vs. Tat−/Sal Fold Change	Tat−/Meth vs. Tat−/Sal FDR BH *p* Value
Gm9456	predicted gene 9456	2.001916	0.000107467
LOC100039183	serine/threonine-protein kinase MARK2-like	2.003197	2.90 × 10^−8^
Gm1574	predicted gene 1574	2.035995	0.002292041
Zfp395	zinc finger protein 395	2.036129	0.001748075
1700110I01Rik	RIKEN cDNA 1700110I01 gene	2.036459	1.13 × 10^−6^
Rarres1	retinoic acid receptor responder (tazarotene induced) 1	2.048939	0.005048167
Fbxl13	F-box and leucine-rich repeat protein 13	2.058912	0.001058074
Anp32a	acidic (leucine-rich) nuclear phosphoprotein 32 family, member A	2.075739	0.038582564
Mt3	metallothionein 3	2.079847	2.92 × 10^−7^
Itprip	inositol 1,4,5-triphosphate receptor interacting protein	2.081287	7.17 × 10^−5^
Wnt3	wingless-related MMTV integration site 3	2.088605	0.000620551
Zfyve1	zinc finger, FYVE domain containing 1	2.093638	6.59 × 10^−9^
Fam89a	family with sequence similarity 89, member A	2.128657	6.71 × 10^−5^
Gm6729	predicted gene 6729	2.129533	1.17 × 10^−7^
Iqub	IQ motif and ubiquitin domain containing	2.132557	0.000114519
Hadha	hydroxyacyl-Coenzyme A dehydrogenase/3-ketoacyl-Coenzyme A thiolase/enoyl-Coenzyme A hydratase (trifunctional protein), alpha subunit	2.143009	7.35 × 10^−9^
Eppk1	epiplakin 1	2.143974	0.001827944
2810410L24Rik	RIKEN cDNA 2810410L24 gene	2.155585	3.74 × 10^−6^
Olfr738	olfactory receptor 738	2.164412	0.022625224
Wdr96	WD repeat domain 96	2.165153	0.002248195
Akap14	A kinase (PRKA) anchor protein 14	2.168142	0.002020088
Ngly1	N-glycanase 1	2.181488	5.73 × 10^−7^
Arl6ip1	ADP-ribosylation factor-like 6 interacting protein 1	2.190554	4.98 × 10^−10^
Htr6	5-hydroxytryptamine (serotonin) receptor 6	2.206588	0.045116169
Tmod4	tropomodulin 4	2.216105	1.53 × 10^−5^
Atp10a	ATPase, class V, type 10A	2.286466	0.002265009
3000002C10Rik	glyceraldehyde-3-phosphate dehydrogenase pseudogene	2.32011	4.51 × 10^−10^
Fbxo33	F-box protein 33	2.339012	6.53 × 10^−9^
Rasd1	RAS, dexamethasone-induced 1	2.364438	0.005352506
5430417L22Rik	RIKEN cDNA 5430417L22 gene	2.370661	1.40 × 10^−5^
Rfx7	regulatory factor X, 7	2.396731	0.000533103
Renbp	renin binding protein	2.400166	0.010819928
A4galt	alpha 1,4-galactosyltransferase	2.620867	0.000456334
Hist4h4	histone cluster 4, H4	3.15377	8.77 × 10^−11^
Tmprss11bnl	transmembrane protease, serine 11b N terminal like	3.158875	5.79 × 10−8
1700027A23Rik	RIKEN cDNA 1700027A23 gene	3.438965	8.85 × 10^−9^
Gm10461	predicted gene 10461	4.734624	2.28 × 10^−10^

**Table 4 viruses-12-00426-t004:** Genes that were significantly changed by Tat, upon the comparison between Tat+/Sal and Tat−/Sal.

Gene Symbol	Gene Name	Tat+/Sal vs. Tat−/Sal Fold Change	FDR BH *p* Value
**Cox7B**	cytochrome c oxidase, subunit XVII assembly protein homolog	−2.327017	0.023942447
**Cbx7**	chromobox homolog 7	−1.947657	0.023942447
**Lingo1**	leucine rich repeat and Ig domain containing 1	−1.916766	0.02866605
**1700071K01Rik**	RIKEN cDNA 1700071K01 gene	−1.760823	0.02866605
**Gm9372**	predicted gene 9372	−1.723841	0.02866605
**D230035N22Rik**	RIKEN cDNA D230035N22 gene	−1.696028	0.023942447
**Slco1a5**	solute carrier organic anion transporter family, member 1a5	−1.68099	0.035075396
**Gm17753**	predicted gene, 17753	−1.679336	0.031019153
**Gm3146**	predicted gene 3146	−1.535604	0.023942447
**Mpp3**	membrane protein, palmitoylated 3 (MAGUK p55 subfamily member 3)	−1.520024	0.03363897
**Sgcg**	sarcoglycan, gamma (dystrophin-associated glycoprotein)	−1.511241	0.034422557
**Bcam**	basal cell adhesion molecule	1.587535	0.036223264

**Table 5 viruses-12-00426-t005:** Transcription factor motifs site/sequence matrix identified in the proximal promoter region of genes upregulated by Meth alone, from the comparison between Tat−/Meth versus Tat−/Sal brains, and of genes upregulated by Tat in the context of Meth, from the comparison between Tat+/Meth versus Tat−/Meth brains. NS = not significant.

Matrix	Factor Name	Sequence	Sites Tat−/Meth vs. Tat−/Sal (MSS)	Sites/Sequences Tat−/Meth vs. Tat−/Sal (CSS)	Sites Tat+/Meth vs. Tat−/Meth (MSS)	Sites/Sequences Tat+/Meth vs. Tat−/Meth (CSS)
**V$CPBP_Q6**	KLF6	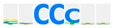	117	5.57	1627	3.86
**V$TATA_01**	TBP-related factors	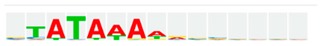	NS	NS	156	2.82
**V$BCL6_Q3_01**	BCL-6 factors	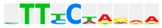	3	3.00	NS	NS
**V$ZNF333_01**	ZNF333	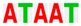	25	2.50	369	1.84
**V$MAZ_Q6_01**	MAZ	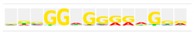	10	2.50	NS	NS
**V$SRY_Q6**	Sox-related factors	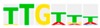	NS	NS	330	1.95
**V$AP1_03**	AP-1 (JUN/FOS)	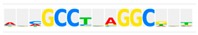	24	2.40	198	2.28
**V$ZFP161_04**	ZFP161	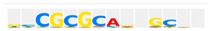	21	2.33	NS	NS
**V$SP1_Q6_01**	Sp1 group	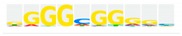	16	2.29	NS	NS
**V$BEN_01**	BEN	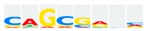	29	2.23	NS	NS
**V$NR3C1_03**	GR-like receptors	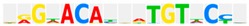	8	2.00	NS	NS
**V$CREBP1_01**	ATF-2 group	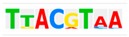	2	2.00	145	2.07
**V$P53_04**	TP53	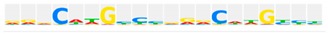	4	2.00	128	1.97
**V$MYOGENIN_Q6_01**	MYOD1	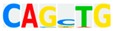	14	2.00	398	2.57
**V$BBX_03**	Bbx	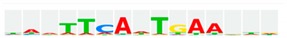	NS	NS	11	1.83

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
