# Peer review of "Systems Biology Analysis of the Antagonizing Effects of HIV-1 Tat Expression in the Brain over Transcriptional Changes Caused by Methamphetamine Sensitization"

_viruses, 2020, doi:10.3390/v12040426_

Round 1

Reviewer 1 Report

The manuscript by Basova and co-authors have performed a systems biology analysis of effects of HIV Tat on Meth-induced transcription changes. The study and analyses are extensive, and the manuscript is well-written, and results clearly presented. The manuscript also provided a novel conclusion that Tat expression in the brain has a low acute transcriptional impact but strongly interacts with Meth sensitization, to modify effects in the global transcriptome. While this conclusion is valid in their systems, and provides some novel perspectives, there is a major issue that needs to be addressed to ensure the conclusion is valid. In their system, Tat expression was induced by doxocycline (Dox) treatment for 7 days. The effects of Dox treatment on blocking Meth-induced transcrtiption was not controlled. The dramatic difference between observed in Tat-/Meth- and Tat-/Meth+ was lost when Tat was induced by Dox , which could result from the 7 days treatment with Dox. A essential control for the conclusion would be a Dox-/Meth+, and Dox+/Meth+, in the absence of Tat. This would exclude the contractions of Dox in the system for a clear and valid conclusion. 

Author Response

Dear Reviewer, we thank you so much for your time. We apologize for not being clear. We did treat ALL animals with Dox. Because this is indeed a very important issue, and it is known that Dox has effects on itself.

In the previous version, we had in Materials and Methods section 2.1 that we analyzed 10 animals containing the GFAP promotor-controlled Tetracycline (Tet)-binding protein (Tat-) and 10 containing both the GFAP promotor-controlled Tet-binding protein AND the (Tetracycline responsive element) TRE promotor-Tat protein transgene (Tat+). Separate, in section 2.2, we had written that all animals were treated with Dox.

In this revised version, we re-wrote the first paragraph of section 2.2 to say:"All mice, 10 containing the GFAP promotor-controlled Tetracycline (Tet)-binding protein (Tat-) and 10 containing both the GFAP promotor-controlled Tet-binding protein and the (Tetracycline responsive element) TRE promotor-Tat protein transgene (Tat+), were treated with a doxycycline regimen (doxycycline hyclate; Sigma, St. Louis, MO, USA) of 100 mg/kg, intraperitoneally, once a day for 7 days (after locomotor testing), beginning in the evening (5 PM) before the Meth acquisition phase". This is labelled with a line on the left side of the paragraph.

We thank you for this comment. We hope we have improved clarity, to highlight the validity of our findings.

Reviewer 2 Report

This study explored the premise that “early exposure to drugs of abuse increase the magnitude of later drug-stimulated responses. A transgenic Tat model of inducible expression is used with four experimental groups with the proper controls. Mice (at 5 pm) were first tested for locomotor function, then one hour later injected with Dox, and then sensitized to Meth or saline for 7 days. After 7 days abstinence, all mice were challenged with a single dose of Meth (or saline?, lines 166-167).

Based on the comprehensive gene expression analyses were performed, the main finding is that Tat alone did not have a large impact on gene expression after the 30 min Meth exposure paradigm and that Tat expression blocked the gene induction seen with Meth alone. The authors say such finding was unexpected and differs from prior work from their lab and others.

The questions below are relate to the rigor of the design and data interpretation. 

-The actual P values should be given in the figures and legends.

-Interesting gene hits shown in the Tables and figures were not among the two genes that were validated by RT-PCR. Why so few? As shown in Figure 7 there are many genes in the sirtuin pathway.

-As the mice were not perfused, could this have confounded the results?

-Why were only male mice used?

-What was the level of tat message and protein in the caudate/putamen after Dox induction? Were there any considerations given to normalization?

Author Response

Dear Reviewer #2, we appreciate your time and very constructive comments.

-The actual P values should be given in the figures and legends.

We have added p values to the legends.

-Interesting gene hits shown in the Tables and figures were not among the two genes that were validated by RT-PCR. Why so few? As shown in Figure 7 there are many genes in the sirtuin pathway.

We have prioritized for showing, genes that remained down regulated in the presence of Tat, and that may have implications to the pathogenesis in humans. However, we did validate a battery of other genes. We have now expanded the supplementary figure with gene validations, to add NOS2, and PPARa, both within the Sirtuin pathway.

-As the mice were not perfused, could this have confounded the results?

We did perfuse the animals. We apologize for not clarifying. We have now added this very important detail to the section 2.4 - Brain Harvest.

-Why were only male mice used?

We used male animals mainly because in females there are the confounders of estrous cycle. We take it as a suggestion to compare males and females controlled for estrous cycle from the point of view of systems biology in the near future. 

-What was the level of tat message and protein in the caudate/putamen after Dox induction? Were there any considerations given to normalization?

We did not measure Tat transcription. Tat expression in these Tat transgenic animals’ total brains was determined to be in the range of 1–5 ng/ml, leading to homogeneous levels of astrocytosis. This published observation was now added to the introduction. However, we did not measure in our caudate specimens. Unfortunately, we no longer have left overs of protein to perform this challenging assay. However, we rely on the results from the group that developed the model, reporting that the range of expression does not affect transcriptional outcomes in significant ways. We did have one outlier mouse in the Tat-Meth group, which we did not exclude from the analysis. The groups containing Tat+ animals exhibited very similar outcomes. This can be appreciated in Figure 1, where individual animals in principal component analysis cluster together. Thus, in addition to internal normalization in the assays, we do not have strong concerns regarding the normalization to Tat.

Round 2

Reviewer 2 Report

The authors have addressed most concerns. However, the rationale about why only male mice were used should be included somewhere in the text.

Author Response

Dear Reviewer #2, 

Thank you so much for your comment. We have added the rationale for female mice in the first paragraph of discussion.

Please, don't hesitate to contact if there are any further comments.

We will also perform a thorough English language revision using our in-house editing group. The revised paper will be promptly uploaded as soon as the editing is ready. We appreciate your time.

M. Cecilia Marcondes.